# Effects of Cognicise-Neurofeedback on Health Locus of Control, Depression, and Quantitative Electroencephalography Alpha Asymmetry in Elderly Women

**DOI:** 10.3390/brainsci11070899

**Published:** 2021-07-07

**Authors:** Heewook Weon, Jieun Yoo, Jumhwa Yu, Miso Park, Haekyoung Son

**Affiliations:** 1Department of Brain and Cognitive Science, Seoul University of Buddhism, Seoul 08559, Korea; soojiwon@hanmail.net; 2College of Social Sciences, Yonsei University, Seoul 03722, Korea; clairejieun@gmail.com; 3Neuroscience Research Institute, Seoul University of Buddhism, Seoul 08559, Korea; jjbory@naver.com (J.Y.); stroda@naver.com (M.P.); 4Department of Nursing, Eulji University, Seongnam-si 13135, Korea

**Keywords:** brain wave, cognition, elderly, neurofeedback, neuroscience

## Abstract

Background: With an increase in the aged population, there is a growing concern regarding the care of the elderly. This study aims to identify effects of cognicise-neurofeedback on health locus of control, depression, and quantitative electroencephalography (QEEG) alpha asymmetry in elderly women. Methods: A quasi-experimental control group pre-test–post-test design was used. Korean women aged 65 years or over at a senior welfare center were randomly allocated to the control (*n* = 12) or experimental (*n* = 19) groups from July to October 2019. The intervention consisted of cognicise (exercise with intensified cognitive activity) and neurofeedback twice a week for 10 weeks. The locus of control and depression were measured via self-reported questionnaires. QEEG alpha asymmetry was measured using BrainMaster. Results: Depression significantly decreased in the experimental group (t = 4.113, *p* = 0.001), while internality in the locus of control significantly decreased in the control group (t = 3.023, *p* = 0.012). On the other hand, QEEG alpha asymmetry index differences in F3-F4 between the pre-test and post-test were not significant in the experimental group (t = 0.491, *p* = 0.629) or control group (t = 0.413, *p* = 0.678). Conclusions: Due to the coronavirus disease (COVID-19) pandemic, contact between the elderly and healthcare workers in the clinical practice field has become more restricted. These findings can help decrease negative emotions among elderly women in the community based on an integrated neuroscientific approach.

## 1. Introduction

### 1.1. Background of Study

The surge in the elderly population is a worldwide phenomenon resulting from increased life expectancy and reduced birth rate [1]. In South Korea, the elderly population has sharply increased at a significantly faster rate than in other advanced countries. This has created an aged society, with a proportion of the elderly population aged 65 years or more exceeding 14% in 2018. South Korea is expected to become a super-aged society, with an aging rate of over 20% in 2025 [2]. In most countries, the average life expectancy of women exceeds that of men. In South Korea, the life expectancy of a 65-year-old survivor in 2018 was 22.8 years for women, which was longer than 18.7 years for men. Compared to the Organization for Economic Co-operation and Development average, the life expectancy of elderly women in South Korea was higher by 1.5 years, whereas that of elderly males was higher by 0.5 years [2].

In general, elderly women in South Korea have long life expectancies, with a higher prevalence of chronic diseases and a lower level of perceived health and physical activity than those of elderly men, resulting in higher levels of depression than other age groups or elderly men [3,4]. Physical activity, subjective health perception, and health locus of control, depression, cognition, and social support have been recognized as variables that affect quality of life in the elderly [5,6,7].

In this regard, health locus of control refers to an individual’s belief that they have control over their own health or that their health is controlled by others or by chance or fate. Locus of control is divided into three corresponding areas: internal health locus of control (IHLC), powerful others’ health locus of control (PHLC), and chance health locus of control (CHLC), and the dominant areas differ in each individual [8]. In other words, locus of control is the general expectation of what affects the outcome of an action, and those who expect their own actions to control their health have an internal locus of control, whereas those who believe that external forces, such as chance or powerful others, affect them have an external locus of control. Health locus of control is known to be an important factor in successful aging because it affects the attitude toward one’s own aging, such as satisfaction, general happiness, and quality of life [9].

Depression, commonly experienced by the elderly, threatens mental health and, along with senile diseases that cause cognitive decline, such as dementia, lowers life satisfaction in later life, increases negative self-perception, and lowers their life expectancy [10,11]. Depression in the elderly requires a more fundamental intervention and preventive approach, as the treatment service referral rate is lower than the prevalence rate [10]. The prevalence of dementia in elderly women (95% confidence interval) was found to be 8.66% (6.02, 11.30), which was higher than 4.36% (2.49, 6.23) in elderly men. The prevalence increases dramatically with age and becomes significantly higher in elderly women than in elderly men over 80 years [12]. Cognitive decline, including senile dementia, tends to worsen from the onset of the disease, places physical, emotional, and financial burdens on the elderly and their families, and leads to a vicious cycle that worsens quality of life in the elderly through depression and lowered self-esteem [11].

In recent years, the brain mapping function has enabled quantitative analysis of brain waves, along with visual judgment and provided useful brain-related information based on the frequency of brain waves [13]. Quantitative electroencephalography (QEEG) is a type of EEG that facilitates quantitative analysis of brain waves [13]. This examines differences in depression and cognitive decline as QEEG is a quantitative analysis of the frequency of brain waves that is more objective than self-reported questionnaires. In particular, alpha waves are closely associated with cognitive aspects and are activated and involved in learning that requires complex brain functions [14], which can be used to examine the effectiveness of cognitive interventions.

Unlike traditional models of depression, which focus on the neurochemical dysfunctions that accompany its phenomenological manifestations, alternative models emphasize the electroencephalographic correlates for diagnostic and prognostic perspectives [15]. A previous study revealed that hypoactivity of the left frontal areas would be associated with depression [16,17,18]. Furthermore, alpha activity of EEG is commonly linked with lower metabolic activation, and the left hypoactivity is associated with relatively higher right than left frontal alpha power [19]. The right frontal lobe was more active in patients with depression than in normal people and examined the differences in emotional or behavioral disposition by EEG based on the asymmetric characteristics due to functional differentiation of the left and right hemispheres of the frontal lobe [16,17,20,21,22,23]. EEG-based neurofeedback in depression trains patients to shift the left and right frontal asymmetry in order to rebalance alpha activation levels [19].

Our society needs to pay attention to and take action towards helping elderly women plan and prepare for healthy aging and lead a successful later life. In particular, there is increasing interest in improving the neurocognitive health and physical function of elderly women. This study aims to determine the effects of cognicise-neurofeedback on health locus of control, depression, and QEEG alpha asymmetry in elderly women and prepare a foundation for ways to improve neurocognitive health and physical function in elderly women in a super-aged society.

### 1.2. Purpose of Study

The specific objectives of the study were as follows: (1) to investigate the level of health locus of control, depression, and QEEG alpha asymmetry in elderly women; (2) to analyze the effects of cognicise-neurofeedback on these variables in elderly women.

## 2. Methods

### 2.1. Research Design

This study was designed as a quasi-experimental control group pre-test–post-test study to analyze the effects of cognicise-neurofeedback on health locus of control, depression, and QEEG alpha asymmetry in elderly women (Figure 1).

### 2.2. Research Participants

The participants of this study were selected by convenience sampling of elderly women aged 65 years or older, registered at a senior welfare center located in Gyeonggi-do, South Korea, who understood the purpose of this study and voluntarily agreed to participate. Participants had sufficiently good physical activity and behavior to participate in this study. Based on a previous study verifying the effectiveness of integrated interventions, such as neurofeedback training and brain exercise [19,24,25,26], a total of 35 elderly women participated in this study (Figure 2). Participants were randomly allocated to the experimental group or control group considering the intervention and data collecting period. In the experimental group, four participants dropped out during intervention and data collection for re-employment and personal reasons, etc. As a result, the data of 19 participants in the experimental group and 12 in the control group were used for the final analysis.

### 2.3. Research Procedures

#### 2.3.1. Preliminary Assessment

In the preliminary assessment, general information, such as age, religion, health status, economic status, residential type, concerns over dementia, and the characteristics associated with participation in interventions, such as cognitive function and social support for exercise, were investigated. Cognitive function was determined using the Korean version of the Mini-Mental State Examination for Dementia Screening (K-MMSE-DS), developed by Kim et al. [27]. The K-MMSE-DS is a clinical tool widely used in screening for dementia that determines the risk of cognitive decline [28]. Social support for exercise was measured using the translated and modified version [29] of the Social Support Questionnaire (SSQ) developed by Sallis et al. [30]. The SSQ consists of seven questions about how much social support the participant receives from people around them, such as family, friends, neighbors, relatives, co-workers, and exercise center instructors. For each question, the participants were asked to answer using a five-point scale, from “strongly disagree” (1 point) to “strongly agree” (5 points). The higher the total score, the higher the level of social support related to exercise. In a previous study, the Cronbach’s α of the SSQ was 0.89 [29], and it was 0.935 in this study.

#### 2.3.2. Intervention: Cognicise-Neurofeedback

Cognicise-neurofeedback as used in this study was developed by the researchers [31] based on previous studies that verified the effectiveness of integrated interventions using neurofeedback [24,32]. The intervention used, based on the study by Weon et al. [31], was provided as 10 min of cognicise and 20 min of neurofeedback per session for 10 weeks (20 sessions in total) in a separate space in the senior welfare center for the participants in the experimental group. Considering that the frequency and duration of the intervention was 2 days a week and 20 min per session in a previous meta-analysis of an intervention integrated with exercise provided for the elderly [5], it was organized at a level that was reasonable for elderly women. Cognicise involves performing simple physical activities (e.g., walking in place in a designated space of 50 cm × 50 cm, etc.), while engaging in a cognitive activity of remembering and speaking the words provided in advance by one syllable. For the neurofeedback, the participants were trained according to the protocol of the training mode for brain function enhancement, focusing on relaxation, concentration, and memory with the neurofeedback training device while sitting on a chair. The relaxation, concentration, and memory training used for neurofeedback consisted of challenges such as memorizing the colors or shapes of images displayed on the screen for a limited time or remembering and matching the locations of the same images among multiple images displayed for 1 s (Figure 3). For the neurofeedback training, a qualified expert was used, who had a Board-Certified Neurofeedback license conducted with Neurobrain (Neuro21, Korea), which was developed in the Korea Brain Science Research Institute. Neurobrain trained the left and right EEG in FP1 and FP2 of the prefrontal cortex simultaneously using two measuring electrodes and used FPz as the ground electrode and the left earlobe as the reference electrode. Theta (4–8 Hz) and beta (16–20 Hz) waves were inhibited and a low beta (12–15 Hz) wave was reinforced. Those EEGs were measured using the 33120A Function Generator (HP, USA), and reliability was demonstrated with the 984A attenuator (Kikusui, Japan) at 0.945 (*p* < 0.01) [33].

#### 2.3.3. Outcome Assessment

##### Health Locus of Control

Health locus of control was measured by the Multidimensional Health Locus of Control (MHLC), developed by Wallston et al. [8]. This tool was used to measure the location of an individual’s perceived control over health, and it consists of 18 items with three sub-sections of six items each on IHLC (internality), PHLC (powerful others externality), and CHLC (chance externality). Each sub-section was independent, and the scores were not summed. Each item was answered on a 6-point scale from “strongly disagree” (1 point) to “strongly agree” (6 points). The higher the total score, the higher the level of health locus of control. The Cronbach’s α of the original tool was 0.77 for IHLC, 0.62 for PHLC, and 0.69 for CHLC [8], whereas it was 0.533 for IHLC, 0.470 for PHLC, and 0.600 for CHLC in this study.

##### Depression

Depression was measured using the Korean version of the Center for Epidemiological Studies and Depressive Symptomatology (K-CES-D) [34], a tool used for self-reported screening of depression, translated and standardized by Cho and Kim [35]. This tool consists of a total of 20 self-report items about the condition of the participant over the past week. Each item is answered on a four-point scale from “Very Rarely” = one day or less per week (0 point) to “almost always” = five days or more per week (3 points). The higher the total score, the higher the level of depression, weighted by frequency of occurrence during the past week. The possible range of scores is 0 to 60, with the cut-off at 16. The Cronbach’s α of the tool was 0.90 [35], whereas it was 0.906 in this study.

##### Quantitative Electroencephalography

Quantitative electroencephalography (QEEG) data were collected by a researcher with Board Certification in Neurofeedback using the BrainMaster Discovery (BRAINMASTER TECHNOLOGIES, INC., Bedford, OH, USA). The QEEG data were collected by measuring the alpha waves (8–12 Hz) from F3 and F4, based on the international 10–20 system [36]. The EEG signal was transformed using Cz montage (Cz is the common reference site) [18,36,37] and by quantifying with the NeuroGuide software (Applied Neuroscience Inc., St. Petersburg, FL, USA) to examine the frontal alpha asymmetry. The participants were seated in comfortable chairs in front of a computer monitor, where EEG was displayed in real time and data were collected for 12 min during wakeful relaxation. The brain’s electrical activity was displayed on a monitor in the form of brain waves. The participants were encouraged to maintain a relaxed state, closing their eyes and moving as little as possible to reduce artifacts caused by muscle tension. The QEEG data were analyzed after performing artifact editing with NeuroGuide and transforming the data into a frequency band using the fast Fourier transform algorithm. The EEG data were collected with a reliability of 0.90 or higher. As for the English letters and numbers in F3/F4, F indicates the frontal lobe, odd numbers of electrodes indicate the left hemisphere, and even numbers of electrodes indicate the right hemisphere [38].

### 2.4. Data Collection

This study was conducted after obtaining approval (No. 27004121AN01-201906-HR-057-02) from the Institutional Review Board of Seoul University of Buddhism and from the relevant senior welfare center from July to October 2019. The researcher provided a detailed explanation of the study that was easy to understand using the structured guide and allowed the participants to fully consider prior to giving consent to participation. As for data collection, three trained researchers helped the participants read and complete the questionnaire and collected the QEEG data. The experimental group participated in 20 sessions of cognicise-neurofeedback in total, and the control group, who had not participated in the cognicise-neurofeedback, were given the opportunity to participate if they desired after the study was completed. The researcher explained the QEEG measurement results to the participants.

### 2.5. Data Analysis

The collected data were analyzed using the SPSS/WIN 20.0 program. Variables, such as general characteristics of participants, characteristics associated with participation in intervention, health locus of control, depression, and QEEG alpha asymmetry of elderly women, were analyzed as descriptive statistics; the correlation between variables was analyzed using Pearson’s correlation coefficient, and the comparison of effects before and after intervention was analyzed by paired *t*-tests. The statistical significance level was set at *p* < 0.05.

## 3. Results

### 3.1. Test of Homogeneity

The general characteristics of the participants are shown in Table 1. The age of the participants ranged from 65 to 82 years, and the mean ages (standard deviation (SD)) of the experimental and control groups were 74.58 (5.34) years and 74.33 (4.03) years, respectively. As for health status, the most common response was “neutral”, with 8 (42.1%) from the experimental group and 5 (41.7%) from the control group. With regard to economic status, 16 (84.2%) and 11 (91.7%) participants in the experimental and control groups, respectively, answered “neutral” or worse. The responses for the residential types included 12 (63.2%) and 6 (50.0%) participants of each group living with their family and 7 (36.8%) and 6 (50.0%) participants of each group living alone. Regarding concerns over dementia, 14 (73.7%) and 8 (66.6%) participants in each group were “very worried” or “worried.” As for cognitive function measured by K-MMSD-DS, the mean (SD) of the experimental group and the control group was 25.78 (3.64) points and 25.92 (2.75) points, respectively. Regarding social support for exercise, the mean (SD) of the two groups was 2.58 (8.39) points and 18.08 (9.15) points, respectively. As shown, the general characteristics and the characteristics associated with participation in the intervention were homogenous in both groups (*p* > 0.05). In addition, as a result of preliminary validation of homogeneity of the experimental group and the control group (Table 2), the quantitative analysis of health locus of control, depression, and frontal alpha asymmetry revealed homogeneity in both groups (*p* > 0.05).

### 3.2. Effects of Cognicise-Neurofeedback

The effects of cognicise-neurofeedback were analyzed (Table 3). There was no significant difference in health locus of control in the experimental group, but internality was significantly reduced in the control group (t = 3.023, *p* = 0.012). Depression was significantly reduced below the cut-off of 16 in participants in the cognicise-neurofeedback group compared to that before intervention (t = 4.113, *p* = 0.001). When comparing the QEEG alpha asymmetry, asymmetry was not significantly reduced in the F3-F4 in the experimental group (t = 0.491, *p* = 0.629) or control group (t = 0.413, *p* = 0.687).

## 4. Discussion

In this study, cognicise-neurofeedback was developed based on the interventional effects of various existing brain education programs integrating cognicise, using both cognitive and physical activities and neurofeedback via brainwave control training [31]. The results of this study revealed a significant decrease in depression of elderly women in the experimental group, compared to the control group.

First, cognicise was performed by engaging in a cognitive activity, such as creating poetry or remembering specific patterns while performing an aerobic physical activity after warming up, including stretching in each session, as described in a previous study [32]. Since cognicise is an intervention that combines cognitive and physical activities, it is necessary to adjust the content and difficulty according to the cognitive and activity levels of the participants, which would result in a difference in terms of effects. In this study, considering that the participants were elderly women who were participating in cognitive training for the first time, the cognicise was composed of movements and cognitive activities that could be easily followed without physical fatigue for a short period of time, similar to a previous study [5]. The average age of elderly women who participated in this study was 74.58 and 74.33 in the experimental and control groups, respectively. As for cognitive function measured by the K-MMSD-DS, a clinically proven screening tool for dementia, the means (SD) of the experimental group and the control group were 25.78 (3.64) points and 25.92 (2.75) points, respectively. According to the distribution of K-MMSE-DS total scores, the participants belonged to the normal group with 25 points or higher, based on a total score of 30 points [39]. Similar to Lee [39], there were no participants in the high-risk group scoring 14 points or lower in this study. Nevertheless, the cognicise seems to have been designed with a difficulty level suitable for both a low-risk group with 20 to 24 points and a moderate-risk group with 15 to 19 points. This could be used as basic data when devising an intervention, considering the level of participation depending on the cognition of elderly women in the community. However, these results appear to reflect the health status of the participants with a decent level of physical function and cognitive level, who could voluntarily register at the senior welfare center in the community and participate in various community programs operated by the senior welfare center.

In this study, the QEEG alpha asymmetry was not significantly reduced in F3-F4 of the frontal lobe associated with depression and neurocognition in elderly women who participated in cognicise-neurofeedback. In Jeong et al. [24], an integrated therapy combining neurofeedback and cognicise for the elderly living alone in the community significantly improved their brain function and quality of life. In particular, depression in the elderly is a factor that influences quality of life, and Choi and Kim [40] reported that an exercise program significantly improved physical fitness, cognitive function, and quality of life, as well as depression in the elderly. Conversely, the results of this study are not in line with the results of previous studies verifying the effectiveness of single or combined interventions, including physical, cognitive, and neurofeedback exercises. For future studies, it is recommended to standardize the composition of time, location, and content of the intervention by providing an integrated intervention with proven effects of improving cognitive function and depression in elderly women with various health issues in the community to ensure effectiveness and accessibility.

Comparing the alpha asymmetry of the elderly women participants in the cognicise-neurofeedback of this study before and after participation showed that it was not significantly reduced in F3-F4 of the frontal *(p* > 0.05). The related brain regions were distributed throughout frontal pole, frontal, antero-temporal, and midline frontal areas. In particular, FP1 and FP2 (Brodmann area 10, 11, 46) in the lateral orbital frontal area were related to cognitive emotional valence, including depression or emotional inhibition [41]. The frontal lobe is a region closely associated with cognitive functions, such as dementia, as it is involved in thinking activities, attention, and working memory. For example, F3 and F4 (Brodmann area 8, 9, 46) were related to short-term memory and F7 and F8 was related to working memory [41]. The decrease in frontal alpha asymmetry indicates an improvement in the cognition of the participants [42,43]. In particular, alpha waves of 8–12 Hz predominantly appear during wakeful relaxation with eyes closed in a normal state. Alpha waves are known to be associated with general alertness and are involved in attention-demanding tasks [14]. In the frontal lobes of elderly women with dementia, slow waves, such as theta and delta waves are activated, as opposed to those of normal elderly women with faster activated alpha waves [44,45]. In general, in the elderly, degradation of brain functions due to aging appears to manifest as a decrease in logical thinking ability, work processing speed, and short-term memory [46]. It may be difficult to clinically distinguish early symptoms of mild dementia from cognitive dysfunction in normal elderly individuals or cognitive dysfunction due to depression. Although it is not easy to distinguish between cognitive decline in normal people and cognitive decline due to dementia, an excellent study has been suggested that can distinguish dementia through QEEG [47,48,49,50]. EEG is a non-invasive and relatively simple method for examining brain functions using electrical signals generated in the process of signal transmission between cranial nerves. Therefore, in future studies, it is suggested to verify the effects of the integrated interventions by analyzing the association between various EEG waves and to standardize the effects and results by applying the intervention to the elderly with various symptoms, such as changes caused by normal aging and degenerative or vascular changes.

### Limitations

This study verified the development and effectiveness of cognicise-neurofeedback based on various existing interventional effects on brain health. However, the insufficient integrated intervention approaches for promoting brain health, cognitive function, and depression in elderly women from previous studies limit direct comparisons with the results of this study. The results should be interpreted considering that the participants of this study were able to voluntarily register at the senior welfare center in the community and participate in various community programs with a decent level of physical and cognitive function. Since this was a quasi-experimental pre-test–post-test control group study, it was an exploratory attempt to investigate the effect of integrated intervention for the elderly registered in one senior welfare center. Due to the limitations of representativeness and the possibility of sampling bias due to convenience sampling, caution needs to be taken in generalizing the results of this study.

## 5. Conclusions and Suggestions

This study investigated the effects of an integrated intervention, focusing on health locus of control, depression, and QEEG alpha asymmetry in F3-F4 by applying cognicise-neurofeedback to elderly women in the community. In this study, a neuroscientific approach was used to verify the effects of cognition-related interventions in elderly women more objectively, based on examining brain waves that are closely associated with cognitive and emotional changes, rather than relying on self-report questionnaires. In particular, EEG is a useful measurement tool that allows objective, non-invasive, and continuous measurement in real time. This study is significant because it provided basic data for an integrated intervention, promoting the brain health of elderly women in the community. Due to the coronavirus disease (COVID-19) pandemic, contact between the elderly and healthcare workers in the clinical practice field has become more restricted. In the future, it is recommended to verify the effectiveness of a community-based integrated intervention and consider a more standardized intervention with improved interventional effects and accessibility that can be provided even for the elderly with various health issues who are unable to participate in programs provided by senior welfare centers.

## Figures and Tables

**Figure 1 brainsci-11-00899-f001:**
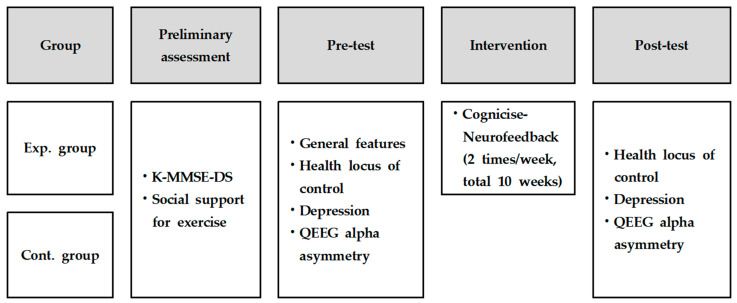
Research design. Exp., experimental; Cont., control; K-MMSE-DS, Korean version of the Mini-Mental State Examination for Dementia Screening; QEEG, Quantitative electroencephalography.

**Figure 2 brainsci-11-00899-f002:**
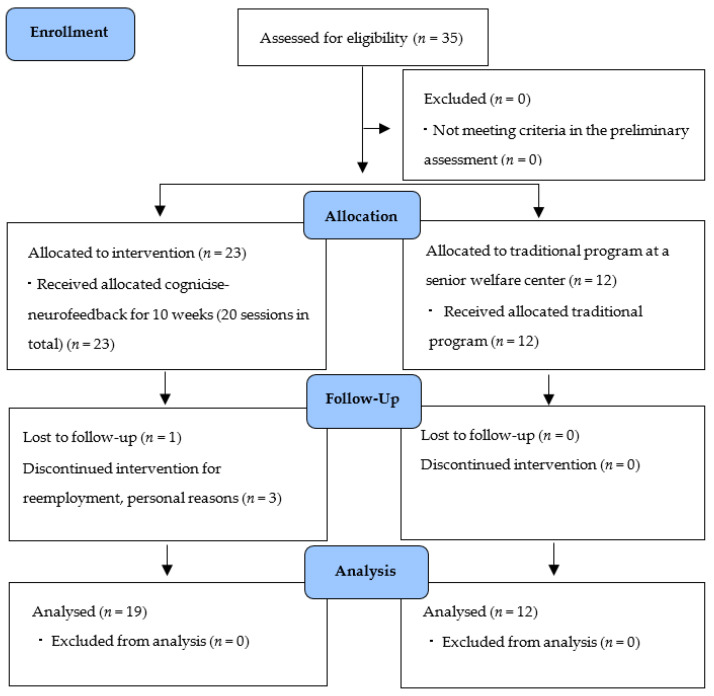
Flow chart.

**Figure 3 brainsci-11-00899-f003:**
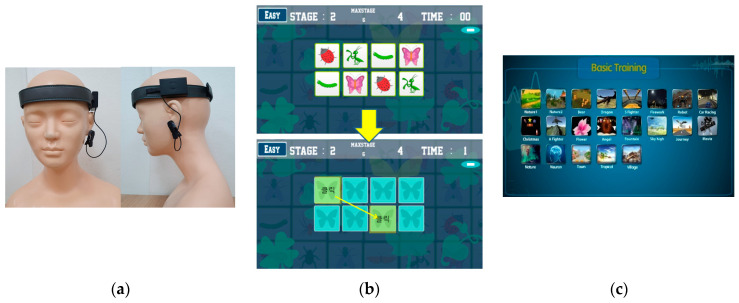
Example of the cognicise-neurofeedback screen. (**a**) wearing device of neurofeedback; (**b**) memorizing and matching the same images; (**c**) categories of basic training screen.

**Table 1 brainsci-11-00899-t001:** Homogeneity of the preliminary assessment.

Characteristics	Categories	Mean (SD)/Frequency (%)		t (*p*)
		Exp. (*n* = 19)	Cont. (*n* = 12)	
Age (years)	≥65 and ≤82	74.58 (5.34)	74.33 (4.03)	0.136 (0.892)
Religion	Christianity	9 (47.4)	2 (16.7)	0.659 (0.515)
	Buddhism	6 (31.6)	4 (33.3)	
	Catholicism	0 (0.0)	5 (41.7)	
	None	4 (21.1)	1 (8.3)	
Health status	Very good	1 (5.3)	1 (8.3)	0.533 (0.598)
	Good	5 (26.3)	4 (33.3)	
	Neutral	8 (42.1)	5 (41.7)	
	Poor	4 (21.1)	1 (8.3)	
	Very Poor	1 (5.3)	1 (8.3)	
Economic status	Very good	0 (0.0)	0 (0.0)	0.746 (0.462)
	Good	3 (15.8)	1 (8.3)	
	Neutral	5 (26.3)	6 (50.0)	
	Poor	4 (21.1)	3 (25.0)	
	Very Poor	7 (36.8)	2 (16.7)	
Residential types	Home with family	12 (63.2)	6 (50.0)	0.705 (0.486)
	Living alone	7 (36.8)	6 (50.0)	
Concerns over dementia	very worried	6 (31.6)	4 (33.3)	0.825 (0.416)
	worried	8 (42.1)	4 (33.3)	
	Neutral	3 (15.8)	0 (0.0)	
	Not worry	1 (5.3)	2 (16.7)	
	Not worry at all	1 (5.3)	2 (16.7)	
Cognitive function	≥15 and ≤30	25.78 (3.64)	25.92 (2.75)	0.112 (0.911)
	≥15 and ≤19	1 (5.3)	0 (0.0)	
	≥20 and ≤24	6 (31.6)	4 (33.3)	
	≥25 and ≤30	12 (63.1)	8 (66.7)	
Social support for exercise	≥7 and ≤35	22.58 (8.39)	18.08 (9.15)	1.404 (0.171)

**Table 2 brainsci-11-00899-t002:** Homogeneity of variables.

Characteristics	Mean (SD)		t (*p*)
	Exp. (*n* = 19)	Cont. (*n* = 12)	
Locus of Control			
Internality	28.37 (4.88)	30.75 (3.17)	1.499 (0.145)
Powerful others’ externality	24.68 (4.81)	23.83 (5.65)	0.079 (0.938)
Chance externality	27.53 (4.72)	28.17 (4.95)	0.361 (0.721)
Depression	25.95 (13.53)	19.50 (10.93)	1.387 (0.176)
Frontal alpha asymmetry	5.23 (36.58)	6.82 (53.10)	0.099 (0.922)

**Table 3 brainsci-11-00899-t003:** Comparison of variables between pre-test and post-test.

Characteristics	Exp. (*n* = 19)			Cont. (*n* = 12)		
	Mean (SD)		t (*p*)	Mean (SD)		t (*p*)
	Pretest	Posttest		Pretest	Posttest	
Locus of Control						
Internality	28.37 (4.88)	28.04 (3.38)	0.254 (0.802)	30.75 (3.17)	26.01 (3.88)	3.023 (0.012) *
Powerful others’ externality	24.68 (4.81)	23.47 (6.32)	0.883 (0.389)	23.83 (5.65)	23.58 (3.29)	0.864 (0.406)
Chance externality	27.53 (4.72)	25.47 (6.32)	1.186 (0.251)	28.17 (4.95)	26.67 (3.39)	1.017 (0.331)
Depression	25.95 (13.53)	14.88 (7.93)	4.113 (0.001) *	19.50 (10.93)	14.45 (9.77)	1.929 (0.080)
Frontal alpha asymmetry	5.23 (36.58)	−0.07 (19.76)	0.491 (0.629)	6.82 (53.10)	0.15 (16.34)	0.413 (0.687)

* *p* < 0.05.

## Data Availability

The data presented in this study are available on request from the corresponding author. The data are not publicly available due to restrictions eg privacy or ethical.

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
