# Peer review of "Effects of Cognicise-Neurofeedback on Health Locus of Control, Depression, and Quantitative Electroencephalography Alpha Asymmetry in Elderly Women"

_brainsci, 2021, doi:10.3390/brainsci11070899_

Round 1

Reviewer 1 Report

There is an inadequate review of literature.  Line 82-85 cites only one study.  There should be references to a large number of studies (e.g., by Richard Davidson among others).  It is especially vital that this complex literature cite that asymmetries are primarily found between F3 and F4 when using a Cz reference (not linked ears).  Line 112 again cites only 1 reference that is related to neurofeedback plus brain exercise, but there are more studies that may be cited for neurofeedback with depression (e.g.,  Baehr, Rosenfeld, & Baehr, 2001;Hammond, 2001, 2005; Hammond & Baehr, 2009; Paquette et al., 2009; Choi et al., 2010;  Cheon et al., 2015). 

I definitely disagree on line 76-77 that QEEG "allows even non-professionals to easily examine" it.  Line 149-150:  There is inadequate detail about what the specific parameters of the neurofeedback protocol were and with what equipment.  What was being reinforced and inhibited and in what areas?  Line 185:  Why was qEEG data only examined for alpha?  There could have been other interesting findings in other frequency bands.  Line 250 on.  Re-analyze the data after re-referencing (which is easy in NeuroGuide) to a Cz reference and examining absolute power to see if your findings are compatible and congruent with other publications that treated depression focusing on changing the F3-F4 alpha asymmetry.  There were 21 asymmetry analyses run, of which only 6 were significant (and none were run with a Cz reference).  Thus I question the appropriateness of saying (line 295) that the alpha asymmetry was significantly reduced.  I find inaccurate reporting on lines 311-312 compared with the table, with the reported changes at F4-F8, F4-Fz, Fp1-F8, and Fp2-F8 seeming non-significant.  

Lines 322-324:  There are excellent QEEG studies demonstrating the ability to discriminate dementia that should be cited (e.g., Koenig et al., 2005; Moretti et al., 2004; Gawel et al, 2009; Boerman et al., 1992; Stam et al., 1996; Nobili et al., 1999)

Author Response

Thank you for your valuable feedback for the improvement of this article. Upon general review of the manuscript, current study was added to the reference list and further explanations to enhance readers’ understanding. Thank you again for such detailed feedback. If anything else is required with regard to this, I will be glad to consider the same.

Reviewer 2 Report

Cognitive deficits are core symptoms of depression. This study aims to investigate whether neurofeedback (NF) training can improve depression and qEEG alpha asymmetry in elderly women and elevate neurocognitive health performance. The subject of the research is very important and timely, considering the effect of the pandemic on the elderly population. While this study should be published to encourage the scientific community and society to reconsider the approach to improve healthy aging. Unfortunately, I find the study has insufficient evidence and lacks raw data representation. I believe the authors should rearrange the data presentation to make the study more convincing for the readers. I have some major concerns, which I believe need to be considered before publishing this article.
Concerns:

  • Authors should define the affected brain regions with EEG data and show the raw data of the alpha power in affected brain regions.
  • Authors should consider including the differences in the eyes-closed alpha wave in all participants with a cumulative data representation. In that figure, it should further visualize the effects of different brain regions.
  • Authors should consider to include raw data for the baseline EEG alpha asymmetry.
  • All the data appeals to have a correlation analysis between the depression and brain region-specific alpha wave.
  • I think authors should consider including a proposed model to explain their findings and expected mechanism. This will improve the understanding of the readers.
  • Authors should reconsider the heading, as the same wordings have been repeated multiple times in the abstract and discussion as a phrase.
  • Authors should check the writing; they have many repeating sentences throughout the article.

Author Response

(The authors gave the same response as above.)

Round 2

Reviewer 1 Report

The authors have rationalized using a linked ears reference.  Yes, it is commonly used, but the best research finding a frontal alpha asymmetry between F3 and F4 used a Cz reference.  A Cz reference can be created in the NeuroGuide program.  I know because I use it to further examine linked ears findings.

Author Response

Thank you for your valuable input. Upon general review of the manuscript, we re-analyzed the data after re-referencing to a Cz and examining the F3-F4 alpha asymmetry. Furthermore, current study was also added to the reference list and the contents were revised accordingly. If anything else is required with regard to this, I will be glad to consider the same. Thank you again for your valuable feedback for the improvement of this article.

Reviewer 2 Report

I have no further comments. I believe the authors improved the article and it can be published now.

Author Response

Thank you again for your valuable feedback for the improvement of this article.